# Grade 3–4 Immune-Related Adverse Events Induced by Immune Checkpoint Inhibitors in Non-Small-Cell Lung Cancer (NSCLC) Patients Are Correlated with Better Outcome: A Real-Life Observational Study

**DOI:** 10.3390/cancers14163878

**Published:** 2022-08-11

**Authors:** Nadia Guezour, Ghassen Soussi, Solenn Brosseau, Baptiste Abbar, Charles Naltet, Charles Vauchier, Nicolas Poté, Lorry Hachon, Céline Namour, Antoine Khalil, Jean Trédaniel, Gérard Zalcman, Valérie Gounant

**Affiliations:** 1Thoracic Oncology Department-Early Phases Unit CIC-1425 Inserm, Institut du Cancer AP-HP.Nord, Hôpital Bichat-Claude Bernard, 46 Rue Henri Huchard, 75018 Paris, France; 2Université Paris Cité, 75018 Paris, France; 3Immunotoxicity Multidisciplinary Board PATIO, Institut du Cancer AP-HP.Nord, Paris, France; 4Pulmonogy and Thoracic Oncology Department, Hôpital Saint-Joseph, 75014 Paris, France; 5Pathology Department, Institut du Cancer AP-HP.Nord, Hôpital Bichat-Claude Bernard, 46 Rue Henri Huchard, 75018 Paris, France; 6Pharmacy Department, Institut du Cancer AP-HP.Nord, Hôpital Bichat-Claude Bernard, 46 Rue Henri Huchard, 75018 Paris, France; 7Radiology Department, Institut du Cancer AP-HP.Nord, Hôpital Bichat-Claude Bernard, 46 Rue Henri Huchard, 75018 Paris, France

**Keywords:** pembrolizumab, nivolumab, ipilimumab, immunotherapy, immune checkpoint inhibitors, immune-related adverse events, prognosis, non-small-cell lung cancer

## Abstract

**Simple Summary:**

Immune checkpoint inhibitors (ICIs) recently became a standard treatment for advanced non-small-cell lung cancers (NSCLCs). Immune-related adverse events (irAEs) could occur in 10 to 80% of treated patients but were reported to associate with a better prognosis in clinical trials. However, the prognostic role of Grade 3–4 irAEs, occurring in 2 to 18% of cases, has not been specifically addressed in a real-life setting yet. In this observational study, we highlighted an association between high-grade irAEs and better outcomes in advanced NSCLC patients who received ICI treatment. Actually, a significantly longer overall survival was observed in patients with high-grade irAEs compared to the no-irAEs group. This observation thus suggests a direct link between anti-tumor efficacy and the level of immune activation leading to high-grade irAEs.

**Abstract:**

Background: Immune checkpoint inhibitors (ICIs) have been a major advance in treating non-small-cell lung cancer (NSCLC). Programmed cell death protein-1/programmed death-ligand 1 blockade enhances immune function, mediating anti-tumor activity, yet causing immune-related adverse events (irAEs). We investigated the prognostic role of Grade 3–4 irAEs on overall survival (OS). Methods: This observational study recruited advanced NSCLC patients who received ICIs at Bichat-Claude Bernard University Hospital and in a community hospital, Saint-Joseph Foundation (Paris), between 1 January 2016 and 31 December 2019. Immunotherapy as a single-agent or double-drug combination was applied in the first and later lines. Univariable and multivariable analyses were instrumental in evaluating the prognostic impact of irAEs. Results: Overall, 201 consecutive ICI-treated patients were enrolled. High-grade irAEs (Grades 3–4) occurred in 36 patients (17.9%), including 11 (30.5%) cases of pneumonitis, 8 (22.2%) of colitis, 4 (11.1%) hepatic, 3 (8.3%) dermatological, 2 (5.5%) neurological events, and 2 cases (5.5%) of poly-arthralgia. The median OS was 10.4 ± 1.36 months (95% CI:7.7–13.1), being significantly higher in patients with high-grade irAEs than those without, 27.8 months vs. 8.1 months, respectively (HR = 2.5; *p* < 0.0001). Multivariable analysis revealed an independent association between high-grade irAEs and longer OS (HR = 0.29, 95% CI: 0.2–0.6, *p* < 0.0001). Conclusions: Our real-life study confirms that high-grade irAEs predict longer OS in advanced NSCLC.

## 1. Introduction

Immune checkpoint inhibitors (ICIs) have significantly prolonged long-term survival in a patient subset with advanced non-small-cell lung cancer (NSCLC) although real-world studies including fewer selected patients only reported moderate survival increase with the use of ICIs [1]. Initially developed for the second line [2,3], ICI use in 2022 has become indispensable for treating metastatic NSCLC, mostly in association with platinum-based chemotherapy from the frontline [4,5].

Upon ICI-treated patients’ follow-up, new toxicities occurred. By blocking immune checkpoints, ICIs disrupted immune homeostasis, causing immune-related adverse events (irAEs). In NSCLC patients, depending on the combination used, anti-programmed cell death protein-1 (PD-1) agents, given either alone or in combination with anti-cytotoxic T-lymphocyte antigen 4 (CLTA-4), irAEs were reported to occur in 10–80% [6] of ICI-treated patients. The incidence of Grade 3–4 irAEs with single-agent anti-CTLA-4 ipilimumab at 3 mg/kg dosing (while used at 1 mg/kg in NSCLC), was reported to be 26% in the randomized Phase 3 trial CheckMate 067 [7], while the two nivolumab registration trials for second-line setting, in squamous and non-squamous NSCLC, at 3 mg/kg dosing, reported lower rates from 7 to 10% [2,3], similar to the 13.6% incidence of Grade 3–5 irAEs in the Phase 3 first-line pembrolizumab trial [4]. The highest incidence of irAEs has been reported for the nivolumab plus ipilimumab combination with 33% Grade 3–4 irAEs in the CheckMate-227 trial [8] as compared with the 19% incidence in the nivolumab arm. Such a higher rate of severe irAEs was also reported in the meta-analysis of urological cancers ICI trials [9], which also identified the length of exposure to ICI as a variable associated with an increased incidence of severe irAEs [9], although most of the irAEs occur within the first months of treatment [7,8].

The real-world incidence of irAEs is likely higher, given that immunotherapy is now more commonly employed by less experienced centers not yet involved in the first ICI clinical trials. Most irAEs tend to be mild and self-limiting, but severe cases (Grades 3 or 4) were reported in 2–18% of ICI-treated patients, with potential life-threatening events [10]. Yet, the drug-related death rate turned out to be much lower than with chemotherapy.

Several studies suggested that patients with irAEs may experience greater clinical long-term benefits. This association between irAEs and improved clinical outcomes was first described in melanoma patients [11]. Several NSCLC studies reported similar outcomes, without specifically focusing on Grade 3–4 irAEs. Yet, these studies presented several limitations, including a short-time follow-up resulting in survival analyses with too many censures. The current observational, retrospective, and long-term follow-up study conducted in two academic hospitals sought to investigate whether there was a significant association between Grade 3–4 irAEs and overall survival (OS) in ICI-treated advanced-stage NSCLC patients.

## 2. Materials and Methods

### 2.1. Study Design, Objectives, and Participants

This observational, retrospective study involving patients with advanced NSCLC (stage IIIB not amenable to radiotherapy and Stage IV), was conducted in two Parisian hospitals (Bichat-Claude Bernard Hospital; Saint Joseph Foundation) from 1 January 2016 to 31 December 2019 to assess the frequency of irAEs and their prognosis impact in a real-world setting. Files from all consecutive patients with advanced histologically proven NSCLC treated with either pembrolizumab, nivolumab, or ipilimumab, whether alone or in combination, and regardless of treatment line, were retrieved from electronic patient records. Patients with small-cell lung cancer were excluded. According to the French regulatory rules, all new patients received an information leaflet stating that clinical, biological, and progression data would be anonymized before analysis. They were reminded that they could oppose their data utilization at any time. The study was approved by the Institutional Review Board of the French learned society for respiratory medicine (*Société de Pneumologie de Langue Française* (CEPRO); number #2021-039).

### 2.2. Categorization and Definition of irAEs

Toxicity was assessed using the Common Terminology Criteria for Adverse Events (CTCAE v.5) criteria, and iRAEs were classified based on the affected system: respiratory (pneumonitis), gastrointestinal (colitis), dermatological, hepatic, articular, neurological, endocrine, and other irAEs such as asthenia, ocular irAEs, and pancreatitis. Adverse drug reactions (ADR) were not considered immune-related events since driven by alternative mechanisms.

### 2.3. Data Collection and Baseline Measurements

Demographic, clinical, and biological data were collected including gender, age at diagnosis, Eastern Cooperative Oncology Group-performance status (ECOG/PS), smoking history, prior autoimmune disease, chronic lung, and cardiovascular diseases, cancer diagnosis date, cancer histology type, stage, tumor PD-L1 expression status, number and type of metastatic localizations, leukocytes, neutrophils, lactate dehydrogenase (LDH), lymphocyte counts, and Lung Immune Prognostic Index (LIPI) score [12]. Collected treatment data included drug names and treatment lines, first and last ICI administration dates, concurrent treatments like corticosteroids (>10 mg/day), antibiotics, and proton pump inhibitors during the previous month or first three months of ICI treatment, or for corticosteroids (>10 mg/day) only during the previous month of ICI treatment. Grade 3 and 4 IrAEs’ toxicities were recorded.

### 2.4. Study Endpoints

Overall Survival (OS) was the primary endpoint, and time to new treatment (TTNT) was the secondary endpoint.

### 2.5. Statistical Analysis

Anonymized data were analyzed using IBM SPSS Statistics for Windows, Version 25.0 (IBM Corp., Armonk, NY, USA). Descriptive statistics were expressed as frequencies and percentages for categorical variables, and as mean, standard deviation, median, and interquartile range (IQR) for continuous variables. Pairwise between-group comparisons were performed using Pearson’s Chi-squared or Fisher’s exact tests for discrete variables, and Student’s t or Mann–Whitney U tests for continuous variables. OS was calculated from the date of first ICI cycle initiation to the date of death from any cause. The time to new treatment (TTNT) was defined as the time from the first ICI cycle initiation date to the next-line systemic therapy initiation date. The follow-up duration was evaluated using the reverse Kaplan–Meier method. In univariable analysis, OS and TTNT were estimated using the Kaplan–Meier estimator, with the log-rank test applied to assess between-group differences. The data cut-off date was set to 25 October 2020. Patients who had not met the study outcome at data cut-off, meaning death or new treatment initiation, were censored at their last clinical visit date. The multivariable analysis applied backward stepwise Cox regression modeling, including variables with *p*-value ≤ 0.20 from the univariable analysis. All hypothesis tests were two-tailed, with *p*-values < 0.05 indicative of statistical significance. Graphics were computed using RStudio Version 2022.02.3+492, with the “Prairie Trillium” Release for Windows (Integrated Development Environment for R. RStudio, PBC, Boston, MA, USA).

## 3. Results

### 3.1. Cohort Characteristics

From 1 January 2016 to 31 December 2019, 201 patients received ICIs for advanced NSCLC, including 162 (81%) in Bichat-Claude Bernard Hospital and 39 (19%) in Saint Joseph Foundation. Patient clinical and demographic characteristics are provided in Table 1. Overall, 132 patients (66%) were men, and ECOG-PS was ≤1 in 121 patients (60%), with 40% exhibiting PS ≥ 2. The median age was 64 years (IQR: 57–70 years). Overall, 190 (94.5%) patients were current or former smokers with only 11 (5.5%) having never smoked.

The histological subtype was non-squamous NSCLC in 120 patients (60%) and squamous NSCLC in 55 (27%). Overall, 176 (87%) patients were Stage IV and 13% Stage III with a contraindication for thoracic radiation or surgery, with 99 patients (49%) displaying more than 3 metastatic sites.

Considering ICI treatment, 189 (94%) patients received an anti-PD-1 antibody (n = 189), 12 (6%) a nivolumab-ipilimumab association (n = 12), while 140 (70%) were pre-treated with chemotherapy (n = 140). According to treatment recommendations at this period, an ICI-chemotherapy combination was given to none.

Treatment with proton pump inhibitors, antibiotics, or corticosteroids (>10 mg/day) during the previous month or three first months of ICIs was registered in 73 (36%) and 79 (39%) respectively, while corticosteroids treatment during the previous month (>10 mg/day) was identified in 33 patients (16%). LIPI scores were calculated for 200 patients, considering the neutrophil to lymphocyte ratio (NLR) and LDH level, as reported in the literature [12]. Three prognostic categories were identified, including good, intermediate, and poor [12], accounting for 56, 101, and 43 patients, respectively.

### 3.2. Grade 3–4 irAEs

Among the 201 patients, 36 (17.9%) experienced serious irAEs (Grade 3–4), including 11 (30.5%) cases of pneumonitis, 8 (22.2%) of colitis, 4 (11.1%) hepatic toxicities, 3 (8.3%) dermatological events, 2 (5.5%) neurological toxicities, 2 (5.5%) articular events, 1 (2.7%) adrenal insufficiency, and 5 (13%) various other irAEs (1 Grade 3 asthenia, 1 cystoïd macular edema, 1 dry eye syndrome, 1 exocrine pancreatic insufficiency, and 1 pancreatitis) (Figure 1). The clinical and tumor characteristics of patients with Grade 3–4 irAEs did not significantly differ from those without high-grade irAEs (Appendix A) except for proton pump inhibitor intake and single-agent versus combination ICI.

All respiratory Grade 3–4 irAEs (n = 11) were suspected based on radiologic features or mild symptoms (mostly New York Heart Association class ≥2 dyspnea on exertion, hypoxemia, and need for oxygen supplementation, fever, or cough). They were confirmed using diagnostic bronchoscopy whenever possible (n = 7), with a bronchoalveolar lavage revealing a lymphocytic (mainly CD8+) alveolitis supporting the diagnosis, without any infectious concurrent agent. Two of these patients required transient oxygen supply, while four (36.5%) displayed various hypoxemia levels (from mild to severe).

Colitis cases were all diagnosed after biopsy during colonoscopy or recto-sigmoido-colonoscopy, to differentiate microscopic colitis (macroscopic normal colonoscopy with histological lymphocytic infiltration) treated with oral gastro-resistant budesonide (entocort: EC) from erosive colitis requiring systemic corticosteroids. Among the four hepatic irAEs, there were two Grade 3 and one Grade 4 cytolysis cases, the latter being associated with bilirubine increase and histologically confirmed as immune cholangitis. All required oral corticosteroids at >1 mg/Kg/day for several weeks with permanent ICI interruption.

Dermatological events, mostly rashes without any bullous disease, benefited from a complete clinical specialist assessment, with punch skin biopsy revealing lymphocytic infiltration.

The two reported neurologic Grade 3–4 irAEs included one case of aseptic meningitis, revealed by a headache with normal brain magnetic resonance imaging (MRI) and lumbar puncture, and one case of hypophysitis, revealed by asthenia and biological hypocorticism, which was confirmed using MRI.

The two patients experiencing Grade 3–4 articular toxicity were eventually diagnosed with rhizomelic pseudo-polyarthritis, with excellent and fast clinical response to low-dose oral corticosteroids.

Except for the completely asymptomatic Grade 4 pancreatitis (Grade 4 increase in lipasemia), ICIs were stopped after serious irAEs. However, two attempts at ICI resumption were made, the first in a patient with Grade 3 pneumonitis, at 8 months following nivolumab interruption and 6 months of oral corticosteroid intake, with the very same drug and dosage. There was no pneumonitis relapse but fast tumor progression after three nivolumab infusions. The second attempt concerned a patient exhibiting a Grade 3 dry eye syndrome, two years after interrupting nivolumab because of tumor progression. It did not result in any irAE recurrence, and a partial response was obtained, lasting nine months until further progression.

### 3.3. Correlation between Serious irAEs and Better Outcomes

With a median follow-up of 31.6 months (95%CI: 30.3–33.0), the median OS for the entire population was 10.4 months (95%CI: 7.7–13.1) (Figure 2). Patients with Grade 3 or 4 irAEs displayed a significantly longer OS compared with the no-irAEs group, 27.8 months (95%CI: 17.0–38.7) vs. 8.1 (95% CI: 5.9–10.4), with a statistically significant difference in univariable analysis (hazard ratio (HR) = 2.5; 95%CI:1.6–4.1, *p* < 0.0001) (Figure 3), and multivariable analysis adjusted for gender, PS, antibiotic or systemic corticosteroid intake before/at ICI initiation, number of metastatic sites, brain or liver metastases, and LIPI score (adj. HR = 3.0; 95%CI: 1.8–5.1, *p* < 0.0001).

In univariable analysis, female gender, PS ≤ 1, absence of antibiotic or corticosteroid intake, fewer than three metastatic sites, higher PD-L1 tumor expression, lower TNM stage, absence of liver or brain metastases, and lowest LIPI score were all significantly associated with a longer OS.

Other variables, not yet primarily considered, were associated with shorter survival rates. Notably, an adverse influence was found for having received antibiotic therapy, with a median OS of 7.2 months (95%CI: 4.9–9.5) vs. 13.5 months (95%CI: 9.1–17.9) for those who did not (HR = 1.6; 95%CI: 1.2–2.3, *p* = 0.004) (Figure 4A). In the multivariable analysis with (Appendix A) or without PDL1 (Table 2) the antibiotics’ impact remained significant (adj.HR = 1.6; 95%CI: 1.1–2.3, *p* = 0.001). Similarly, corticosteroid intake resulted in a median OS of 4.7 months (95%CI: 2.0–7.4) vs. 12.0 (95%CI: 8.5–15.4) in those without corticosteroids (HR = 2.3; 95%CI: 1.5–6.5, *p* < 0.0001) (Figure 4B). The corticosteroids’ impact remained significant in the multivariable analysis with adj. HR = 2.0, 95% CI: 1.2–3.2, *p* = 0.007. Intermediate and poor LIPI score classes predicted a shorter OS. The median OSs for low vs. intermediate vs. high LIPI scores were 15.3 months (95%CI: 9.8–20.8), 9.9 (95%CI: 6.6–13.2), and 5.8 (95%CI: 0.1–12.1), respectively (Figure 4C). In multivariable analysis (Table 2), LIPI score remained associated with a significantly worse influence on OS (*p* global < 0.001), with adj. HR _(intermediate vs. low)_ = 2.1 (95%CI: 1.3–3.2, *p* = 0.001), and adj. HR _(high vs. intermediate)_ = 1.8 (95%CI: 1.1–3.1, *p* = 0.017).

### 3.4. Time to Next Treatment (TTNT)

Median TTNT in the whole population was 3.9 months (95% CI: 1.3–6.5) (Appendix A). Among patients who experienced Grade 3–4 irAEs, TTNT was 27.5 months, while being only 2.1 months among those without Grade 3–4 irAEs (Figure 5). The univariable analysis yielded a statistically significant difference with an HR of 2.4 (95% CI: 1.6–3.7, *p* < 0.0001). Female gender, PS ≤ 1, absence of corticosteroid intake, less than three metastatic sites, absence of liver metastases, and lower LIPI score were significantly correlated with longer median TTNT in both univariable and multivariable analyses (Appendix A). On the contrary, PD-L1 status failed to predict TTNT (Appendix A).

The occurrence of irAEs of any grade was confirmed to be an independent predictor for longer TTNT in the multivariable analysis adjusted for gender, PS score, antibiotic or corticosteroid intake, number of metastases, presence of liver metastasis, and LIPI score (adj. HR _(absence vs. occurrence G3–4 irAEs)_ = 2.4; 95%CI 1.6–3.8, *p* < 0.0001).

## 4. Discussion

To our knowledge, this study is one of the first to specifically focus on the correlation between Grade 3–4 irAEs and better outcomes in advanced NSCLC patients. Several publications reported a longer OS and progression-free survival (PFS) for patients with irAEs of any grade, but rarely mentioned the specific adverse prognostic value for severe events [13]. We selected only Grade 3–4 irAEs to limit the information bias in a retrospective study setting. In this real-life study, 36 patients (17.9%) were identified from the electronic hospital files as exhibiting severe irAEs. A Phase 1 study (BMS 936558-ONO-4538) similarly collected 14% of Grade 3–4 irAEs with second-line nivolumab in advanced NSCLC [14]. Another study including 296 cases of NSCLC, prostate cancers, colorectal cancers, melanomas, or renal cell carcinomas, likewise documented 14% of Grade 3–4 irAEs [4]. Eun et al. reported an 18.2% rate of severe toxicities in a study seeking to identify risk factors of irAEs [15]. Sung et al. analyzed the specific impact of severe irAEs in a prospective 97-patient cohort, those with Grade ≥ 3 irAEs (only six/97) were more likely to display treatment response than those with no or only low-grade irAEs (68% vs. 20%, *p* = 0.023) [16].

Considering the bias of retrospective studies in PFS analysis, we included the TTNT parameter following ICI treatment or death, whatever occurred first, as an imperfect readout for PFS. TTNT has been largely discussed in the literature for being inherently biased [17]. Indeed, this parameter is unable to differentiate the reasons for initiating a subsequent treatment line (ICI toxicity or tumor progression). Moreover, the exact date at which treatment is resumed could depend on the patient’s general condition, the time needed to recover from an ICI-related AE, or the patient’s will. While imperfectly reflecting PFS, TTNT turned out to be positively affected by Grade 3–4 irAEs, in univariable and multivariable analyses, supporting the prognostic effect of such events. Additionally, antibiotic and corticosteroid intake and poor LIPI score significantly affected TTNT, thus supporting its reliability as a PFS surrogate in real-life retrospective studies. We confirmed the higher prevalence of serious irAEs for the anti-PD-1 plus anti-CTLA-4 combination, with six Grade 3–4 irAEs occurring in the 12 patients treated with this combination (50%) vs. 30 (14.9%) in those undergoing a single anti-PD1 treatment (n = 189). This rate is in line with data from the large Phase 3 trial CheckMate 227, reporting a 32.8% rate of serious irAEs [8]. We observed 11 severe immune-related pneumonitis events (30.5% of all irAEs and 5.5% of all ICI-treated patients). Although thyroid dysfunction was common, thyroid irAEs were typically mild (Grade 1–2).

Post hoc analyses involving prospective Phase 3 clinical trials similarly suggested an association between irAEs of any grade and better outcomes. At the 2021 ASCO conference, a large pooled study on the three Impower trials (Impower 130, 132, and 150) in which chemotherapy with atezolizumab was compared with conventional chemotherapy alone in 1557 patients, reported a 48% (n = 753) rate of irAEs of any grade and only 11% (n = 174) of Grade 3–4. A significantly better outcome was revealed in the irAEs group with a higher response rate (61.1% vs. 37.2% in the control group) and longer median OS (25.7 months vs. 13 months, HR = 0.69, 95%CI: 0.6–0.78) [18]. While our series reported irAEs in ICI-treated patients without chemotherapy, one could speculate that the immunosuppressive effect of chemotherapy could have counteracted the intensity of ICIs’ immune-mediated lung toxicity, with a slightly lower rate of Grade 3–4 irAEs in the Impower study. Conversely, it is also possible that a fraction of the irAEs from this pooled study, including pneumonitis or hepatitis, could have been chemotherapy-related. The analysis of 1150 patients from the Checkmate 9-LA trial, who received either nivolumab, ipilimumab plus two cycles of chemotherapy, or chemotherapy alone, revealed a longer OS in patients who experienced irAEs of any grade (2-year OS: 54% vs. 38% in the control group), re-enforcing our own findings [19].

The mechanism underlying the beneficial effects of severe irAEs on clinical outcomes remains unclear. A first obvious hypothesis is that patients responding to ICIs may be more likely to develop irAEs since they could display a better immune reactivity. PD-L1 activation leads to T-cell apoptosis and immunosuppressive cytokine secretion (IL-10). Disruption of such immune tolerance induced by anti-PD-L1/PD-1 could partly explain the irAEs mechanism. Since PD-L1 expression is validated as a major predictor of a successful outcome, we could easily figure out the correlation between ICI efficacy and toxicity. Yet, it is now admitted that gut microbiota do play a major role in cancer development, but also in the treatment response, including immunotherapy [20]. Chaput et al. described an association between a bacteria, e.g., *Faecalibacterium*, and both tumor response and colitis in ipilimumab-treated patients [21]. This role of microbiota might explain the adverse effects we observed with antibiotics in the first months of ICI treatment (HR 1.7, 95% CI: 1.1–2.6, *p* = 0.021) [22,23].

According to Bomze et al., tumors with a high tumor mutational burden (TMB) were prone to be more susceptible to result in irAEs, owing to a higher neo-antigen charge, with immune hyper-reactivation [24]. Such association between irAEs and better outcomes may be accounted for by an underlying neoantigenic potential stemming from a high TMB. Yet, we were unable to confirm this hypothesis given that TMB assays are not routinely used in France.

The development of immune-mediated toxicities was followed by immunotherapy discontinuation in most study cases in line with the literature, which recommends systemic therapy discontinuation for Grade 3–4 toxicity, along with moderate- to high-dose corticosteroid therapy [25]. A meta-analysis on tociluzimab [26] likely enabled the continuation of immunotherapy with remarkable efficacy, which still needs to be confirmed by controlled prospective studies. Only one of our patients received an anti-Janus Kinase 2 treatment for rhizomelic pseudo-polyarthritis, with good clinical efficacy but without resuming ICIs.

In this study, we did not analyze the correlation between the length of exposure and the risk of irAEs since it was not a pre-specified endpoint. However, as reported in the literature, in our study a large majority of irAEs occurred within the first 6 months of ICI treatment, and very few beyond. Moreover, in patients with Grade 3–4 irAEs, ICI was interrupted and re-challenged very rarely, leading to a short duration of treatment in such patients, which could obscure an eventual analysis of the length of exposure to ICI.

Our study exhibits several limitations, the most prominent being its retrospective design. Yet, such design made it possible to show the impact of antibiotic or corticosteroid intake at immunotherapy initiation, exerting an adverse prognostic influence. This supports the reliability of our methodology, even if we cannot ensure that all out-of-hospital prescriptions were collected pertaining to the month preceding the diagnosis. Another limitation was that the irAEs frequency and their prognosis value upon chemo-immunotherapy were not studied. This was due to the patients’ being included prior to the chemo-immunotherapy area, whereas this treatment has now become standard care for patients with tumor PD-L1 expression <50%. Finally, the limited sample size could be considered an important limitation although our results are corroborated by several other studies [18,19]. However, we performed a Cox multivariate rigorous analysis to exclude as many as possible biases linked to confounders, which is the strength of our study, since actually, the occurrence of Grade 3–4 irAEs did predict independently and significantly longer survival in such analysis.

## 5. Conclusions

Immunotherapy has become a standard treatment for advanced NSCLC. In this study, the occurrence of high-grade irAEs was associated with better outcomes in a real-life setting, suggesting a link between anti-tumor efficacy and immune activation extent. To clarify the indication of re-challenge, a better understanding of the precise mechanisms underlying immune-mediated toxicities and their predictive effects on tumor control is necessary. However, this might not always be useful, on account of the long-term immunotherapy effects in patients with disease control, even without resuming ICI.

## Figures and Tables

**Figure 1 cancers-14-03878-f001:**
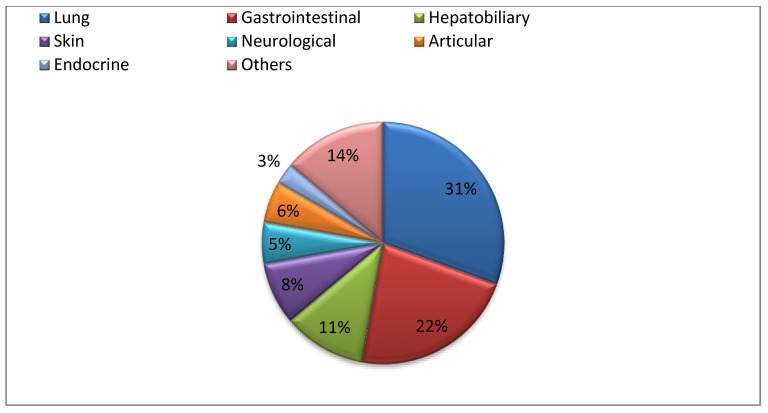
Type and incidence of Grade 3–4 irAEs. Grade 3–4 irAEs occurring in the series of 201 consecutive patients receiving ICI from 1 January 2016 to 31 December 2019 are summarized in a pie chart, showing that the most prevalent Grade 3–4 irAEs are respiratory (31%) and gastrointestinal (22%) while endocrine Grade 3–4 irAEs are rare (3%), linked to non-thyroid irAEs.

**Figure 2 cancers-14-03878-f002:**
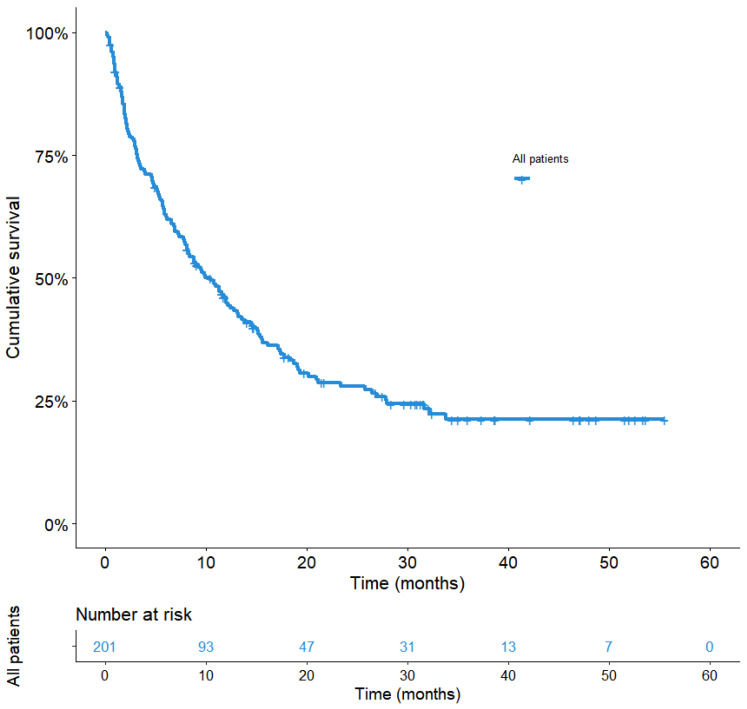
Overall survival in the entire population. Overall survival curve from day 1 of ICI, according to the Kaplan–Meier method is shown for the 201 accrued patients. Median OS for the whole series was 10.4 months (95%CI: 7.7–13.1).

**Figure 3 cancers-14-03878-f003:**
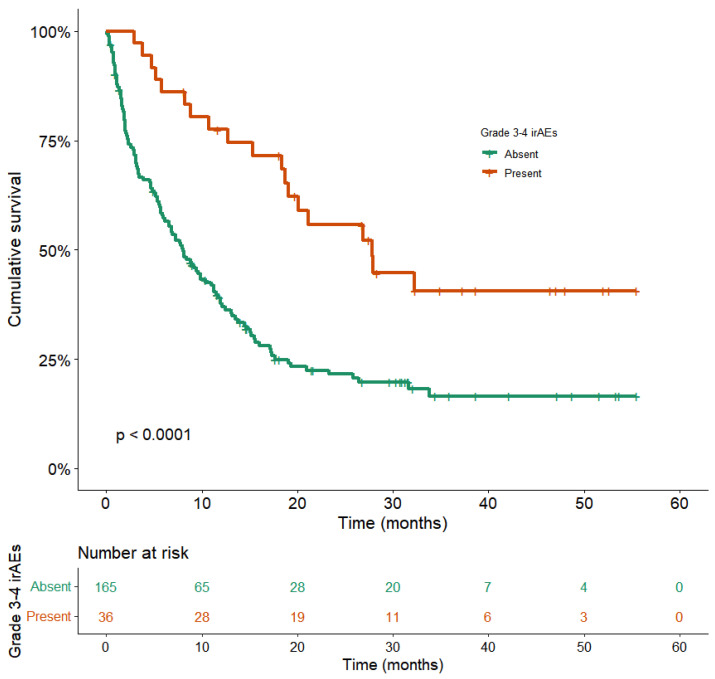
Overall survival according to the incidence of Grade 3–4 irAEs. Kaplan–Meier overall survival plots according to the incidence of Grade 3–4 irAEs are shown. Orange OS curve: patients with Grade 3–4 irAEs (n = 165), median OS: 27.8 months (95%CI: 17.0–38.7). Green OS curve: patients without Grade 3–4 irAEs (n = 36), median OS: 8.1 (95% CI: 5.9–10.4). (*p* < 0.0001, log-rank test).

**Figure 4 cancers-14-03878-f004:**
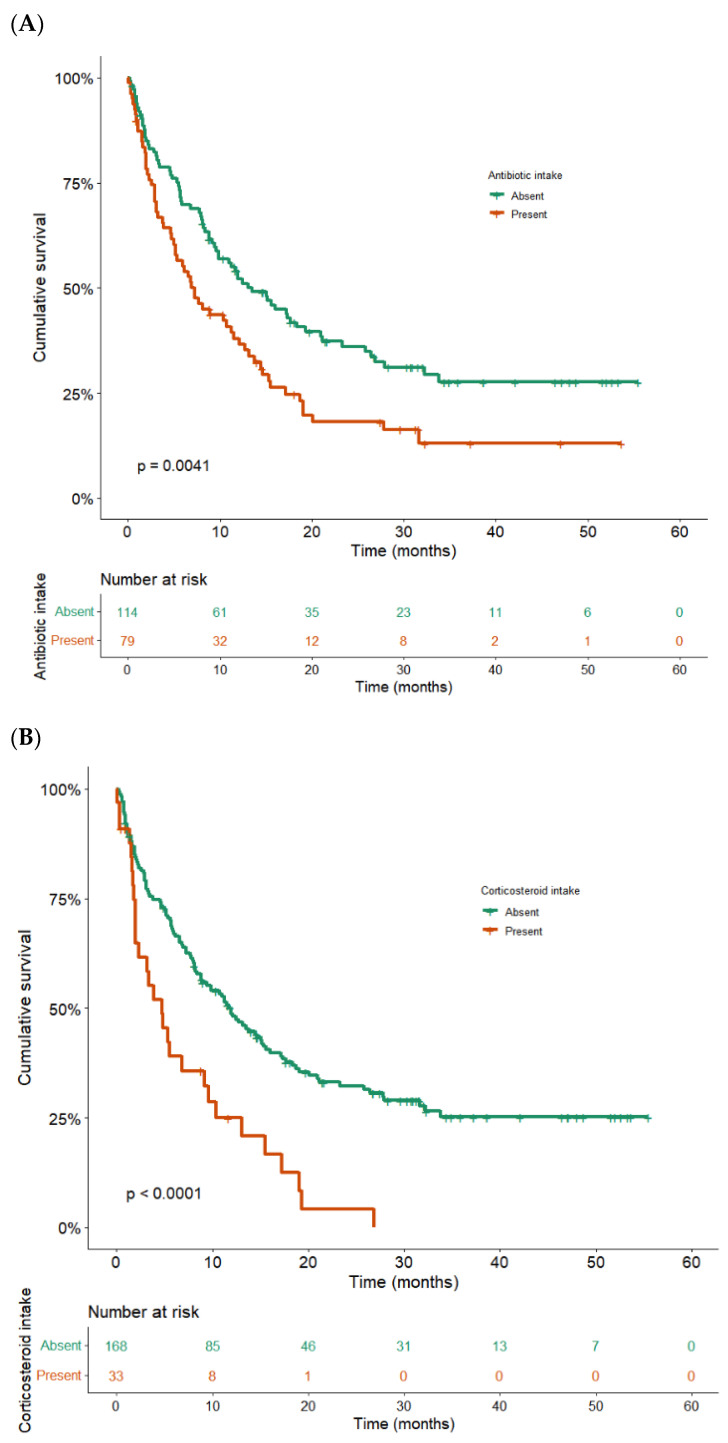
Overall survival according to treatment with antibiotics (**A**), corticosteroids (**B**), and LIPI score (**C**). (**A**) Kaplan–Meier overall survival curves according to treatment with antibiotics the month preceding ICI initiation or the first three months of ICI treatment (n = 79), or not (n = 114). Median OS in case of antibiotics intake history was 7.2 months (95%CI: 4.9–9.5) vs. 13.5 months (95%CI: 9.1–17.9) in patients without identified antibiotic intake (*p* = 0.004, log-rank test). (**B**) Kaplan–Meier overall survival curves according to treatment with corticosteroids within the previous month before ICI initiation (n = 33) or not (n = 168). Median OS in case of corticosteroid treatment the month preceding ICI was 4.7 months (95%CI: 2.0–7.4) vs. 12.0 (95%CI: 8.5–15.4) in patients who did not receive corticosteroids before ICI (*p* < 0.0001, log-rank test). (**C**) Kaplan–Meier overall survival curves according to LIPI score (low vs. intermediate vs. high score). The median OS for low, intermediate, and high LIPI scores were 15.3 months (95%CI: 9.8–20.8), 9.9 (95%CI: 6.6–13.2), and 5.8 (95%CI: 0.1–12.1), respectively (*p* = 0.017, log-rank test).

**Figure 5 cancers-14-03878-f005:**
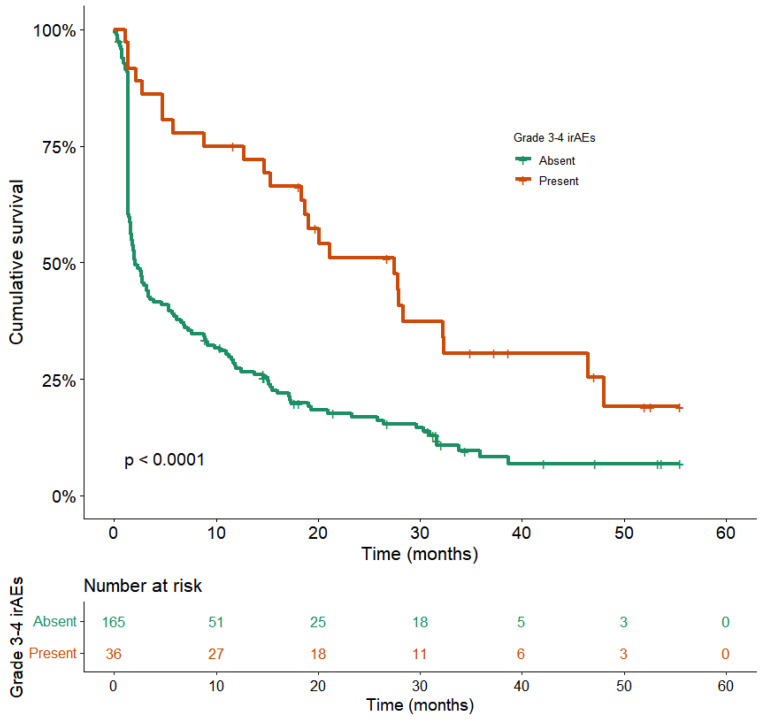
TTNT according to the occurrence of Grade 3–4 irAE. Time to next treatment (TTNT) Kaplan–Meier curves, according to the occurrence of Grade 3–4 irAEs. Median TNT was 27.5 months (95%CI: 16.2–38.8) in the case of Grade 3–4 irAEs, but only 2.1 months (95%CI: 1.1–3.1) in absence of Grade 3–4 irAEs (*p* < 0.0001, Log-Rank test).

**Table 1 cancers-14-03878-t001:** Baseline characteristics.

Characteristics (n = 201)	n (%)
Age at introduction of ICI (years)	
Mean	63
Median (IQR)	64 (70–57)
Gender	
Male	132 (66%)
Female	69 (34%)
Hospital	
Bichat Hospital	162 (81%)
Saint Joseph Foundation	39 (19%)
Tobacco status	
Smoker/Former smoker	190 (92%)
Non-smoker	11 (6%)
Histology	
Adenocarcinoma	120 (60%)
Squamous cell carcinoma	55 (27%)
Others	26 (13%)
Performance status	
0	27 (13%)
1	94 (47%)
2	69 (34%)
3	11 (6%)
Number of metastatic sites	
<3	102 (51%)
≥3	99 (49%)
Brain metastases	
Present	136 (68%)
Absent	65 (32%)
Liver metastases	
Present	33 (16%)
Absent	168 (84%)
Stage at diagnosis/Stage at introduction of ICIs	
Stage III	44/24 (22%/12%)
Stage IV	144/176 (72%/87%)
Other	13/1 (6%/1%)
History	
Chronic respiratory disease	61 (30%)
Cardiovascular disease	101 (50%)
Treatment with proton pump inhibitors, antibiotics, corticosteroids *	73/79/33 (36%/39%/16%)
Treatment line	
ICI on the first line	61 (30%)
Pretreatment with chemotherapy (≥ second line)	140 (70%)
ICI type	
Nivolumab	138 (69%)
Pembrolizumab	51 (25%)
Nivolumab + Ipilimumab	12 (6%)
PDL1 rate	
<1%	76 (38%)
1–50%	37 (18%)
50–75%	47 (23%) **

* During the month preceding and/or the three first months following the initiation of ICI treatment. ** PDL1 missing data in 41 patients.

**Table 2 cancers-14-03878-t002:** Multivariable analysis by Cox proportional hazards for overall survival (OS).

Multivariable Analysis			
Variables	aHR	95% CI	*p*-Value
Gender	1.4	1.0–2.1	0.076
Female			
Male			
PS at ICI initiation	2.2	1.5–3.1	**<0.0001**
0–1			
≥2			
Antibiotic intake *	1.6	1.1–2.3	**0.008**
No			
Yes			
Corticosteroid intake *	2.0	1.2–3.2	**0.007**
No			
Yes			
Number of metastatic sites	1.5	1.0–2.3	**0.047**
<3			
≥3			
Brain metastasis	1.4	0.9–2.2	0.097
No			
Yes			
Liver metastasis	1.9	1.2–3.0	**0.006**
No			
Yes			
LIPI score	- 2.1 1.8	1.3–3.2 1.1–3.1	**0.005** **-** **0.001** **0.017**
0			
1			
2			
Grade 3–4 irAEs	3.0	1.8–5.1	**<0.0001**
No			
Yes			

* During the month preceding and/or the three first months following the initiation of ICI treatment. 95% CI: 95% confidence interval; aHR: adjusted hazard ratio; ICI: immune checkpoint inhibitor; irAEs: immune-related adverse events; PS: performance status. PD-L1 was excluded from the modeling procedure due to the important number of missing data. The multivariable analysis included 193 patients with all available data accounting for 137 events. Stage was excluded either from the model since tightly linked to brain, liver, and number of metastases variables.

## Data Availability

The data presented in this study are available on request to the corresponding author.

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
