# Peer review of "Grade 3–4 Immune-Related Adverse Events Induced by Immune Checkpoint Inhibitors in Non-Small-Cell Lung Cancer (NSCLC) Patients Are Correlated with Better Outcome: A Real-Life Observational Study"

_cancers, 2022, doi:10.3390/cancers14163878_

Round 1
Reviewer 1 Report
Your retrospective observational study gives interesting information about irAEs and outcome in patients treated with ICIs. I have some comments:
Introduction
- You state that “Immune checkpoint inhibitors have significantly prolonged long-term survival in a patient subset with advanced non-small-cell lung cancer.” However, I recommend the authors to be still cautious about this statement. Despite the evidence coming from clinical trials, there are several studies conducted in real-word setting that still report low survival increase for patients with NSCLC over the last decade (PMID: 34885238; PMID: 32470017) even after ICIs introduction. Please introduce better this aspect. I guess this is fundamental also for the rational of your study.
- Please check that each sentence got an appropriate citation.
Method.
- First chapter is called "study design objectives and participants". I guess the objectives are stated at the end of introduction section.
- Please define advanced NSCLC: ≥IIIb ?
- What patients did you included? Incident patients with NSCLC? Is the cohort entry date the first record of NSCLC? Or the first record of ICI?
- From the first chapter of the methods, the inclusion and exclusion criteria are not clear.
- How did you assess the ADR is immune-related?
- How was measured the OS? From the start of the treatment with ICIs or from the first NSCLC diagnosis?
- Statistical analysis: I believe log-rank test is not appropriate to evaluate the statistical difference between two survival analysis in observational studies given the possible role of confounders. I suggest removing log-rank test and report only the Cox analysis.
Results
- I suggest authors to add in results chapter two subchapter: “Cohort characteristics” and “irADR”
- Figure 1: Why did you report the prevalence and not the incidence of irADR?
- When antibiotics, corticosteroids were given to the patients? During ICI treatment?
- I guess that grade 3-4 irAEs are treated with corticosteroids. How do you justify the figure 4 panel b in light of the results shown in figure 3.
- Did you tried to perform a stratification of OS based on the ICI line of therapy? I guess, those patients with line 2+ are likely to live less than those with a first-line
- Which variables did you included in the multivariable cox regression?
Author Response
Dear Reviewer 1,
We thank you very much for the review of our paper, which led to clarifications and improvements in the current version. Please find herein the requested answers.
Comments and Suggestions for Authors
Your retrospective observational study gives interesting information about irAEs and outcome in patients treated with ICIs. I have some comments:
Introduction
- You state that “Immune checkpoint inhibitors have significantly prolonged long-term survival in a patient subset with advanced non-small-cell lung cancer.” However, I recommend the authors to be still cautious about this statement. Despite the evidence coming from clinical trials, there are several studies conducted in real-word setting that still report low survival increase for patients with NSCLC over the last decade (PMID: 34885238; PMID: 32470017) even after ICIs introduction. Please introduce better this aspect. I guess this is fundamental also for the rational of your study.
We thank Reviewer 1 for his/her comment. Actually we have now introduced a sentence to modulate out statement on the improvement of long-term survival in NSCLC patients with ICIs page 2:
" Immune checkpoint inhibitors (ICIs) have significantly prolonged long-term survival in a patient subset with advanced non-small-cell lung cancer (NSCLC) although real-world studies including less selected patients only reported moderate survival increase with the use of ICIs (ref: Spini et al. A. Cancers (Basel) 2021 Dec 5;13(23):6129.doi: 10.3390/cancers13236129). ICI use in 2022 has become indispensable for treating metastatic NSCLC, mostly in association with platinum-based chemotherapy from the frontline (refs Gandhi et al. N Engl J Med. 2018 May 31;378(22):2078-2092, Paz-Arez et al. N Engl J Med. 2018 Nov 22;379(21):2040-2051).
- Please check that each sentence got an appropriate citation.
We thank Reviewer 1 and accordingly added the two references dealing with the registration trials of chemo-immunotherapy in both non-squamous and squamous NSCLC.
Method
- First chapter is called "study design, objectives and participants". I guess the objectives are stated at the end of introduction section.
Reviewer 1 is right and we now also added page 2 in this chapter, the following sentence:
" This observational, retrospective study involving patients with advanced NSCLC was conducted in two Parisians hospitals (Bichat-Claude Bernard Hospital; Saint-Joseph Foundation) from January 1, 2016 to December 31, 2019 to assess the frequency of irAEs and their prognosis impact in real-word setting."
- Please define advanced NSCLC: ≥IIIb ?
We modified this sentence, page 3 as followed:
"This observational, retrospective study involving patients with advanced (stage III B not amenable to radiotherapy and stage IV) NSCLC..."
- What patients did you included? Incident patients with NSCLC?
As mentioned in the M&M section all consecutive advanced NSCLC patients treated with ICI during the study period in our both centers were included in our cohort.
Is the cohort entry date the first record of NSCLC? Or the first record of ICI?
The cohort entry date was the first record of ICI from Jan.1 2016, taken arbitrary as the date of the start of such cohort study, since ICI were used in an early access process in France from Autumn 2015, after the presentation of the results of the first phase 3 trials in second-line setting.
- From the first chapter of the methods, the inclusion and exclusion criteria are not clear.
We are not sure to perfectly understand such comment: all consecutive patients with histologically-proven advanced NSCLC and who were treated with ICI in both centers were accrued. We did not accrue patients with small-cell lung cancer. We precise such criteria in the following sentence, page 3:
" Files from all consecutive patients with advanced histologically-proven NSCLC treated..."
- How did you assess the ADR is immune-related?
According to NCIC Common Terminology Criteria for Adverse Events (CTCAE v.5) and to all clinical trials reporting safety data of ICIs, Adverse Drug Reaction were not considered as immune-related since driven different mechanisms. Besides, such ADR are rare with ICI which accordingly do not require any anti-allergic premedication. We now precise such issue in the following sentence:
page 2:
"... and other irAEs like asthenia, ocular irAEs, and pancreatitis. Adverse drug reactions (ADR) were not considered as immune-related events since driven by alternative mechanisms"
- How was measured the OS? From the start of the treatment with ICIs or from the first NSCLC diagnosis?
OS was measured from the start of the treatment with ICIs. It was clearly mentioned in the M& M section, in the statistical paragraph, page 3: " OS was calculated from the date of first ICI cycle initiation"
- Statistical analysis: I believe log-rank test is not appropriate to evaluate the statistical difference between two survival analysis in observational studies given the possible role of confounders. I suggest removing log-rank test and report only the Cox analysis.
It is true that using the KM method (with Log rank testing) does not take into account for possible confounders but our conclusions are drawn from the results of the Cox modelling (and not from the KM results) which does adjust the effect of each variable on the confounding ones. As a matter of fact, we believe that it was totally totally fair to present the univariable analysis in its current aspect. In fact, univariable analysis is used to select candidate variables for the Cox modelling procedure, which is perfectly defendable from a methodological point of view.
Thus, in suppl. Table 2 the results of the univariable Cox modeling with HRs and 95% confidence intervals are presented with the multivariable Cox analysis, while KM analysis was only used for survival curves and to calculate median survivals.
The title of Suppl. Table 2 is actually misleading and has been then amended as followed:
" Univariable and multivariable analyses (Cox proportional hazards) for time to new treatment (TTNT)"
Results
- I suggest authors to add in results chapter two subchapter: “Cohort characteristics” and “irADR”
We thank Reviewer 1 for his/her suggestion and accordingly we separated to paragraphs page 3:
" Results
Cohort characteristics
From January 1, 2016...."
Grade 3-4 irAEs
Amongthe 201 patients, 36 (17.9%) experienced serious...."
- Figure 1: Why did you report the prevalence and not the incidence of irADR?
We are sorry for the use of this misleading terminology. Actually it was "incidence" and we modified such error as followed:
"Figure 1: Type and incidence of grade 3-4 irAEs"
- When antibiotics, corticosteroids were given to the patients? During ICI treatment?
We now mention in the M& M section that we looked in the patients' electronic files the antibiotics intake information within 1 month before ICI initiation and 2 months after, and only before ICI initiation for corticosteroids thus excluding the corticosteroid treatment justified by the occurrence of an irAE during ICI treatment.
" concurrent treatments like antibiotics, and proton pump inhibitors during the previous month or first three months of ICI treatment, or for corticosteroids (>10mg/day) only during the previous month of ICI treatment"
Accordingly we modified the first sentence of the results section as followed:
" Treatment with proton pump inhibitors or antibiotics, during the previous month or three first months of ICIs was registered in 73 (36%), 79 (39%) respectively, while corticosteroids treatment during the previous month (>10mg/day) was identified in 33 patients (16%)."
- I guess that grade 3-4 irAEs are treated with corticosteroids.
Yes but as mentioned above they were not taken into account for the analysis. Only corticosteroids taken before ICI initiation were kept in the analysis
How do you justify the figure 4 panel b in light of the results shown in figure 3.
Taken into account for our answers above, such results are perfectly understandable. Steroids are deleterious before ICI initiation since depleting cell immunity. However, as reported in many studies before, steroids given as treatment to irAEs as the result of (too) efficient immune response are not deleterious for long-term survival, just controlling a (too) efficient immune response.
- Did you tried to perform a stratification of OS based on the ICI line of therapy? I guess, those patients with line 2+ are likely to live less than those with a first-line.
YES we did. And no, the line of treatment did not turn to influence OS upon ICI which could be the result of an unbalance between the two settings in our study, since only 30% of patients during this period of time received front-line ICI, vs. 70% in second or later lines of treatment, reflecting the ICI registrations at this period, since front-line registration occurred later than second-line registration.
Alternatively, when a major / almost complete response upon ICI occurs, whatever the setting, frontline or second-line, such major response could last for a long time, even beyond 5 years, which consists of the novelty of ICI in NSCLC, depending more of the tumor PD-L1 expression (or any other marker such as TMB...) level, rather than obviously on the setting. This is clearly one major difference between ICI and chemotherapy.
- Which variables did you included in the multivariable cox regression?
Variables included in the multivariable Cox regression were pre-defined as variables with p-value<or = 0.2 in univariate Cox OS analysis, as mentioned in the M& M section, in the statistical paragraph, except for PD-L1 in Table 2 since there were too much lacking data, but the analysis with PDL1 is also shown in suppl. Table 3 without any substantial change in the prognosis value of G3-4 irAEs.
On behalf of all authors,
Your sincerely,
Valérie Gounant, MD, MSc

Reviewer 2 Report
The current study aims to evaluate the association between grade 3-4 irAEs and OS in ICI- treated advanced - stage NSCLC patients. The manuscript is well-designed and the authors should be congratulated for the work and for addressing an important topic.
The manuscript is suitable for publication after minor revision:
- Did you find associations between drug exposure and risk of AEs? do you think this may depend on the mechanism of action of ICIs for therapeutic responses and irAEs?
- I recommend expanding the introduction considering how varying rates and AE profiles are dependent on different cancer types, ICI types, clinical settings, and therapy combinations. May consider including this large SR and MA 10.1016/j.eururo.2022.01.028.
Author Response
Dear Reviewer 2,
We thank you very much for the review of our paper, which led to clarifications and improvements in the current version. Please find herein the requested answers.
Comments and Suggestions for Authors
The current study aims to evaluate the association between grade 3-4 irAEs and OS in ICI- treated advanced - stage NSCLC patients. The manuscript is well-designed and the authors should be congratulated for the work and for addressing an important topic.
The manuscript is suitable for publication after minor revision:
- Did you find associations between drug exposure and risk of AEs? do you think this may depend on the mechanism of action of ICIs for therapeutic responses and irAEs?
In this study, we did not analyze the correlation between the length of exposure and the risk of irAEs since it was not a pre-specified endpoint. However, as reported in Literature, in our study a large majority of irAEs occurred within the first 6 months of ICI treatment, and very few beyond. Moreover, in patients with grade 3-4 irAEs, ICI was interrupted and re-challenged very rarely, leading to a short duration of treatment in such patients, which could obscured an eventual analysis of length of exposure to ICI.
We add a sentence in the discussion section to explain such issue page 13.
- I recommend expanding the introduction considering how varying rates and AE profiles are dependent on different cancer types, ICI types, clinical settings, and therapy combinations. May consider including this large SR and MA 10.1016/j.eururo.2022.01.028.
We thank reviewer 2 for this interesting suggestion. Accordingly we added a few sentences with corresponding references in the introduction section to comment on the differences in irAEs frequencies, according to the use of anti-PD-1 anti-CTLA-4, the combo anti-PD-1 plus anti-CTLA-4 or to the combo chemo-immunotherapy. However in this particular series, most of patients received single agent ICI therapy with anti-PD-1 (n=94.1%), first registered in France for second-line setting. Only 6% received anti CTLA-4 ipilimumab, in combination with nivolumab, within clinical trials. We did not comment then on the clinical setting (metastatic vs. preoperative), since here the series only comprises advanced NSCLC (not resectable stage III with contra-indication to radiotherapy and stage IV patients) and it would have increased the length of the article. Besides, the data are scarce yet for neoadjuvant setting in early NSCLC.
We now added the following sentences in the Introduction section page 2:
" The incidence of grade 3-4 irAEs with single agent anti-CTLA-4 ipilimumab at 3mg/kg dosing (while used at 1mg/kg in NSCLC), was reported to be 26% in the randomised phase 3 trial Check-Mate 067 (ref: Lancet oncol. Hodi FS et al. 2018), while the two Nivolumab registration trials for second line setting, in squamous and non-squamous NSCLC, at 3mg/kg dosing, reported lower rates from 7 to 10% [1,2], similar to the 13.6% incidence of grade 3-5 irAEs in the phase 3 first-line pembrolizumab trial (ref: Gandhi, N Engl J Med 2018; 378:2078-2092). The highest incidence of irAEs has been reported for the nivolumab plus ipilimumab combination with 33% grade 3-4 irAEs in the CheckMate-227 trial (ref: Hellmann, N Engl J Med 2019; 381:2020-2031) as compared with 19% incidence in the nivolumab arm. Such higher rate of severe irAEs has also being reported in the meta-analysis of ICI trials in urological cancers (ref: Wu, Eur Urol 2022 Apr;81(4):414-425), which also identified the length of exposure to ICI as variable associated with increased incidence of severe irAEs (ref: Wu, Eur Urol 2022 Apr;81(4):414-425), although most of irAEs occur within the first months of treatment (ref: Hodi, Lancet oncol. 2018)."
On behalf of all authors,
Your sincerely,
Valérie Gounant, MD, MSc
Reviewer 3 Report
I agree with authors opinion that previous studies has many limitations. However, I strongly believe that the present study includes a small number of patients. This is the most important limitation for this study. Moreover, group has many different characteristics which should be taken into consideration, carefully, in order for analysis not be biased.
Furthermore, the differences between males and females could be presented, as well.
Figures: Authors could include more details in legend of thε images
The presentation of results are confused and it is dfficult to follow the message of the paper
Authors used a few number of past publications. They should include more information of references.
Author Response
Dear Reviewer 3,
We thank you very much for the review of our paper, which led to clarifications and improvements in the current version. Please find herein the requested answers.
Comments and Suggestions for Authors
I agree with authors opinion that previous studies has many limitations. However, I strongly believe that the present study includes a small number of patients. This is the most important limitation for this study. Moreover, group has many different characteristics which should be taken into consideration, carefully, in order for analysis not be biased.
We do agree with Reviewer 3 and insisted in the revised version on the relatively small number of patients.
Page 13:
“Finally, the limited sample size could be considered as an important limitation although our results are corroborated by several other studies [18,19]. However, we performed a Cox multivariate rigorous analysis to exclude as much as possible biases linked to confounders, which is the strength of our study, since actually, occurrence grade 3-4 irAEs did predict independently and significantly longer survival in such analysis.”
Furthermore, the differences between males and females could be presented, as well.
Actually, in our analysis, gender did translate into prognosis differences. Still, as it was included in the multivariate models, gender was shown not to impact the independent prognosis role of grade 3-4 irAEs occurrence as evidenced in the suppl. Table 3 (OS multivariate analysis including PD-L1) or in Table 2 (OS multivariate analysis w/o PD-L1) in which Gender is not significant anymore (p=0.076), while G3-4 irAEs occurrence remains highly significant.
Figures: Authors could include more details in legend of the images
OK we did.
Figure 1:
Grade 3-4 irAEs occurring in the series of 201 consecutive patients receiving ICI from January 1, 2016 to December 31, 2019 are summarized in a pie chart, showing that the most prevalent Grade 3-4 irAEs are respiratory (31%) and gastro-intestinal (22%) while endocrine Grade 3-4 irAEs are rare (3%), linked to non-thyroid irAEs.
Figure 2:
Overall survival curve from Day 1 of ICI, according to Kaplan-Meier method is shown for the 201 accrued patients. Median OS for the whole series was 10.4 months (95%CI: 7.7-13.1)
Figure 3:
Kaplan-Meier overall survival plots according to the incidence of Grade irAEs are shown.
Orange OS curve: patients without Grade 3-4 irAEs (n=165), median OS: 27.8 months (95%CI: 17.0-38.7)
Green OS curve: patients with Grade 3-4 irAEs (n=36), median OS: 8.1 (95% CI: 5.9-10.4)
(p< 0.0001, Log-Rank test)
Figure 4:
Fig. 4A: Kaplan-Meier overall survival curves according to treatment with antibiotics the month preceding ICI initiation or the first three months of ICI treatment (n=79), or not (n=114). Median OS in case of antibiotics intake history was 7.2 months (95%CI: 4.9-9.5) vs. 13.5 months (95%CI: 9.1-17.9) in patients without identified antibiotic intake (p=0.004, Log-Rank test)
Fig. 4B: Kaplan-Meier overall survival curves according to treatment with corticosteroids within the previous month before ICI initiation (n=33) or not (n=168). Median OS in case of corticosteroid treatment the month preceding ICI was 4.7 months (95%CI: 2.0-7.4) vs. 12.0 (95%CI: 8.5-15.4) in patients who did not receive corticosteroids before ICI (p<0.0001, Log-rank test)
Fig. 4C: Kaplan-Meier overall survival curves according Overall survival according to LIPI score (low vs. intermediate vs. high score). The median OS for low, intermediate and high LIPI score were 15.3 months (95%CI: 9.8-20.8), 9.9 (95%CI: 6.6-13.2), and 5.8 (95%CI: 0.1-12.1), respectively (p=0.017, Log-Rank test)
Figure 5: Time to next treatment (TTNT) Kaplan-Meier curves, according to the occurrence of GHrade 3-4 irAEs. Median TNT was 27.5 months (95%CI: 16.2-38.8) in case of Grade 3-4 irAEs, but only 2.1 months (95%CI: 1.1-3.1) in absence of Grade 3-4 irAEs (p<0.0001, Log-Rank test)
The presentation of results are confused and it is difficult to follow the message of the paper.
We are sorry that Reviewer 3 found it difficult to follow. However, such opinion seems to be in contradiction with Reviewer 2 who wrote that "The manuscript is well-designed and the authors should be congratulated for the work" and found " the manuscript suitable for publication after minor revision"
We did not know what to precisely change to satisfy Reviewer 3 without more documented remarks. According to Reviewer 1 and 2' comments, we substantially modified the article, especially the presentation of results, and hope it will now satisfy Reviewer 3.
Authors used a few number of past publications. They should include more information of references.
We did increase then number of references as suggested by other the two Reviewers.
We also want to mention that this paper has been extensively edited by a professional English-native medical writer, Dr. Gabrielle Cremer, from Cremer Consulting© and further verified in its revised version.
On behalf of all authors,
Your sincerely,
Valérie Gounant, MD, MSc
Round 2
Reviewer 1 Report
Dear authors,
you provided detailed answers to my questions